# Association between High Waist-to-Height Ratio and Cardiovascular Risk among Adults Sampled by the 2016 Half-Way National Health and Nutrition Survey in Mexico (ENSANUT MC 2016)

**DOI:** 10.3390/nu11061402

**Published:** 2019-06-21

**Authors:** Eduardo Rangel-Baltazar, Lucia Cuevas-Nasu, Teresa Shamah-Levy, Sonia Rodríguez-Ramírez, Ignacio Méndez-Gómez-Humarán, Juan A Rivera

**Affiliations:** 1Evaluation and Surveys Research Center, National Public Health Institute, 62100 Cuernavaca, Mexico; erangelb@insp.mx (E.R.-B.); tshamah@insp.mx (T.S.-L.); 2Center for Nutrition and Health Research, National Public Health Institute, 62100 Cuernavaca, Mexico; scrodrig@insp.mx; 3Center for Research in Mathematics, Aguascalientes Unit, 20259 Aguascalientes, Mexico; imendez@cimat.mx; 4General Director of the National Public Health Institute, 62100 Cuernavaca, Mexico; jrivera@insp.mx

**Keywords:** waist-to-height ratio, abdominal obesity, cardiovascular risk, adults, surveys

## Abstract

Little evidence exists regarding the association between waist-to-height ratio (WHtR) and cardiovascular risk (CVR) factors in Mexican adults. Our study pursued a twofold objective: To describe the association between a high WHtR and CVR indicators among Mexican adults canvassed by the 2016 Half-Way National Health and Nutrition Survey, and to examine the interaction of sex and age on this association. We analyzed data from the adult sample (≥20 years old) and classified in two groups using WHtRs ≥0.5 considered high and low otherwise. The following CVR factors were analyzed: High-total-cholesterol (≥200 mg/dL), low high-density-lipoprotein-cholesterol (HDL-c < 40 mg/dL), high low-density-lipoprotein-cholesterol (LDL-c ≥ 130 mg/dL), high triglycerides (≥150 mg/dL), insulin resistance (IR) (HOMA-IR) (≥2.6), and hypertension (HBP) (≥140/≥90 mmHg). We estimated prevalence ratios (PR) to analyze the association between high WHtRs and CVR indicators. Over 90% of participants had high WHtRs and were at greater risk for dyslipidemias, HBP, and IR compared to those that had low WHtRs. PR for men with high WHtRs were between 1.3 to 2.3 for dyslipidemias, 3.4 for HBP and 7.6 for IR; among women were between 1.8 to 2.4 for dyslipidemias and HBP and 5.9 for IR (*p* < 0.05). A high WHtR is associated with CVR factors in Mexican adults.

## 1. Introduction

Chronic noncommunicable diseases (CNCD) have become a major cause of death in the world. According to the World Health Organization (WHO), in 2015, CNCD accounted for 38 million deaths globally. Cardiovascular diseases (CVD) represented the largest share of these deaths (17.5 million), followed by cancer, respiratory diseases and diabetes [1]. CVD have thus been ranked as the leading cause of death in the world, with their prevalence escalating in both developed and developing countries [2,3]. Obesity, diabetes mellitus, Insulin Resistance (IR), hypercholesterolemia, hypertriglyceridemia, and high blood pressure (HBP) are considered the predominant CVD risk factors [4]. In Mexico, in 2011, premature mortality from CNCD reached 47% in men and 67% in women, with 19% of cases in both sexes relating to CVD [5].

Since obesity constitutes the highest risk factor for CNCD and mortality [6,7], it is imperative to have simple and precise measurements for defining it. Broad meta-analyses [8,9] have associated body mass index (BMI) with mortality. However, although it has long served as the standard obesity measurement in medical practice, BMI does not discriminate between fat and fat-free mass. Abdominal obesity and visceral fat have been directly associated with CVD-related morbi-mortality [10] as well as with a greater likelihood of metabolic abnormalities [11,12]. Other anthropometric measurements have been used as alternatives to the BMI, namely waist circumference (WC), waist-to-hip ratio (WHR), and more recently, waist-to-height ratio (WHtR). While the first two have been strongly associated with CVD events [13] given their relationship with general adiposity, WHtR has demonstrated a direct positive association [14]. While the mechanisms through which WHtR predicts health risks have yet to be firmly established, its direct relationship with abdominal obesity has been suggested as a possible explanation [15].

Important evidence has emanated from the attempts to discern the best anthropometric measurement for predicting CVD risk in adults. In 2008, an analysis of ten scientific reports showed that WHtR outperformed BMI as the best predictor for HBP, diabetes, and dyslipidemias [16]. Additionally, two years later, a systematic review of 78 articles on 14 countries found that WHtR and WC were preferable to BMI for predicting diabetes and CVD [17]. More specifically, based on the receiver operating characteristic (ROC) curve, the authors confirmed that, for medical purposes, the WHtR with the cutoff point set at 0.5 was more suitable than WC as a screening tool. In 2012, a systematic review and meta-analysis including 31 studies on various countries corroborated these findings [18], with subsequent research [19] reaching the same conclusion. However, no studies have analyzed the Mexican population.

In light of the above, the central objective of our work was, on one hand, to describe the association of WHtR with biochemical and clinical cardiovascular risk factors (CVR) among the adults canvassed by the *ENSANUT MC 2016*, and on the other, to examine the effect of sex and age on the WHtR-CVD association.

## 2. Materials and Methods 

This study was cross sectional with information from ENSANUT MC 2016. This is a nationally representative probabilistic survey, was conducted by urban/rural areas and regions including northern, central and southern Mexico as well as Mexico City. It covered 9,479 households across the country with a 77.9% response rate. Details of the survey sampling design have been published elsewhere [20]. On a random basis, we took the blood pressure and drew venous blood specimens from a subsample of the survey representing 60% of the total sample of adults aged ≥20 years. 

### 2.1. Variables of Interest

Waist-to-height ratio (WHtR). Height and WC measurements were taken by trained and standardized personnel [21,22]. Height was measured with a stadiometer (Dynatop, Mexico) with 1-mm precision. WC were taken on umbilical region using a flexible fiberglass tape with 1 mm precision (Gulick, USA), making two measurements without exceeding 5 mm between them. These data served to calculate the WHtRs (WC in cm/height in cm). In conformity with the literature, the risk index was set at ≥0.5 [17].

Sex and age in years. These variables were obtained from the results of the household questionnaire. 

Blood determinations. Blood specimens were obtained in the morning and drawn from the antecubital vein in vacuum tubes (Vacutainer, Beckton Dickinson, Inc., Broken Bow, NE, USA) and centrifuged in situ at 2500 rpm in a portable centrifuge (Hettich, EBA8, Tuttlingen, Germany). The serum was stored in cryotubes covered with aluminum foil to protect them from the light and later preserved in liquid nitrogen until the time of transfer and delivery to the Nutrition Laboratory of the National Institute of Public Health in Cuernavaca, Mexico. All subjects had at least eight hours of fasting. Determinations were measured by an automatic immunoanalyzer (Architect C18200, ABBOTT DIAGNOSTICS, Germany), triglycerides by lipase hydrolysis, total cholesterol by enzymatic digestion and oxidation, and high-density lipoprotein cholesterol (HDL-c) by a direct enzymatic colorimetric method after eliminating the chylomicrons. Interassay variability was 2.3% for total cholesterol, 3.1% for HDL-c, and 4.8% for triglycerides. Low-density lipoprotein cholesterol (LDL-c) was estimated according to the Friedewald equation: LDL-c = total cholesterol − {(triglycerides/5) + HDL-c} [23]. In accordance with the third report of the expert panel on detection, evaluation, and treatment of high cholesterol in adults [24], high total cholesterol was set at ≥200 mg/dL, low HDL-c at <40 mg/dL, high LDL-c at ≥130 mg/d and high triglycerides at ≥150 mg/dL. 

Insulin was measured by chemiluminescence using Access2 equipment (Beckman Coulter, USA) and glucose through the oxidase glucose method using SynchronX equipment (Beckman Coulter, USA). IR was estimated via the homeostatic model proposed by Matthews et al. in 1985 [25], where IR = fasting serum insulin (µU/mL) x fasting plasma glucose (mg/dL)/405, and values ≥2.6 denoted IR [26].

Blood pressure. Based on the technique recommended by the American Heart Association [27], trained personnel took duplicate measures of blood pressure from the dominant arm using a medical-grade digital sphygmomanometer (Omron HEM-907 XL). Hypertension (HBP) was defined as systolic blood pressure ≥140 mmHg and/or diastolic blood pressure ≥90 mmHg [28]. 

### 2.2. Statistical Analysis 

We estimated the means of the anthropometric measurements and the CVR factors by sex. Subsequently, we compared the means of the lipid profile, blood pressure, and IR measurements by WHtR category using Student’s unpaired t test and estimated the prevalences of the CVR factors (dyslipidemias, HBP and IR) by WHtR category and sex. Given that WHtR is considered a risk indicator in the prevalence of the factors under study, the Prevalence ratios (PR) were calculated as measures of association, which can be interpreted as the relative risk that is used when the study design allows us to infer causal effects [29,30]. Finally, we built logistic regression models in order to study the WHtR-related changes in the CVR factors and tested the modifying effects of sex and age on the WHtR through double and triple interactions. We used the STATA v.14.0 statistical package (Collage Station, TX, USA) for all analyses, considering an alpha of 0.05 for principal effects and one of 0.1 for interactions. Analysis focused on characterizing WHtR as a CVD diagnostic criterion and on identifying the contribution of sex and age to the WHtR-CVD relationship. Therefore, we studied only the characteristics of the associations observed in the sample and drew no inferences for the population; analysis did not include either a design adjustment or expansion factors.

All subjects gave their informed consent for inclusion before they participated in the study. The study was conducted in accordance with the Declaration of Helsinki, and the protocol was approved by the Research Ethics and Biosecurity Committees of the National Institute of Public Health in Mexico, (CI:1401, National Institute of Public Health), approved on 19 April 2016. All participants gave written informed consent.

## 3. Results

We analyzed data from 3550 adults ≥20 years of age from the ENSANUT MC 2016 and present their general characteristics in Table 1. The mean ± SD age of the study population was 46.1 ± 16.0 years, significantly higher for men (47.3) than for women (45.4). The prevalence of high WHtR was 91.4% (93.2% for women and 87.9% for men). While men had higher WC than women, they showed lower concentrations of total LDL-c and HDL-c, but higher levels of triglycerides. Men also showed higher blood pressure but lower glucose and insulin levels.

Estimated CVR factors were statistically higher in the high-WHtR category (≥0.5) with HDL-c statistically lower (Table 2).

For both sexes, high WHtR was associated with a higher probability of experiencing dyslipidemias, HBP and IR. Men with a high WHtR were 1.3 (95% CI: 1.1, 1.5) to 2.3 (95% CI: 1.6, 3.5) times more likely to experience dyslipidemias, 3.4 (95% CI: 1.7, 6.7) times more likely to be hypertensive, and 7.6 (95%CI: 3.5, 16.8) times more likely to have IR. Meanwhile, women were 1.8 (95% CI: 1.3, 2.4) to 2.4 (95%CI: 1.8, 3.2) likely to experience dyslipidemias and HBP and 5.9 (95% CI 3.4–10.1) times more likely to have higher IR, both groups were compared with their low-WHtR counterparts (Table 3).

No significance was found in sex and age as modifying effects of WHtR on CVR factors; however, sex and age did prove to be useful in explaining changes in CVR factors, this association being stronger for women (Figure 1). For example, the total cholesterol risk for women peaked at the age of 60, and diminished slowly thereafter, this risk being higher than it was for men (*p* = 0.001). Similar behaviors were observed as regards LDL-c (*p* = 0.001) and triglycerides (*p* = 0.001); it bears mentioning that risk patterns for low HDL-c did not differ by sex across the various age groups (*p* = 0.487). In the case of HBP, the risk for women did not seem to diminish with age compared to men, for whom it showed a slight decrease after age 70 (*p* = 0.010). Similarly, IR demonstrated a stronger association by WHtR category among women, reaching its highest point at the age of 50 (*p* = 0.066). 

Differences emerged in the interaction of sex and age. We found that women were at higher risk for total and high LDL-c from the ages of 50 to 80, while men were at higher risk for triglycerides after the age of 40 compared with their counterparts. No differences were found in the HDL-c nor HBP. The most relevant results pertained to IR, where women showed a higher risk in their 30s, 50s, 60s, and 70s (data not shown).

## 4. Discussion

This study shows how sex and age modifies the positive association between high WHtR (≥ 0.5) and dyslipidemia, HBP and IR, carried out in a national sample of Mexican adults. Other studies have found an association similar between high WHtR and total cholesterol, high triglycerides, low HDL-c, high LDL-c, and HBP [31,32,33]. Another finding was that ≥90% of the participants had high WHtR.

Of all the indicators analyzed in this study, the strongest association was found for IR in both sexes as compared to the remaining CVR factors; this result could be attributable to the cutoff point used (≥2.6), which had high diagnostic sensitivity and low specificity [26]. In addition, it has been documented that a lack of physical activity, independent of WHtR, directly affects IR [34]. Finally, apart from unhealthy weight, a diet rich in fats and high glycemic load are important contributors to IR [35,36]. In Mexico, the diet of the adult population is characterized by excessive consumption of food rich in saturated fats and/or added sugar, providing 26% of total energy intake, with sugary drinks displacing the consumption of vegetables, fruits, legumes, and whole grains [37]. This could explain the combined effects of visceral adiposity and poor eating habits on the high risk of IR in our study. 

With respect to the interactions, differential patterns were observed by sex and age decades. This could be explained, on one hand, by the process of aging accompanied by a sedentary lifestyle directly associated with IR [38]; on the other hand, in spite of a distribution of grease preferably ginecoid in women [39,40], aging is characterized by a higher internalization of visceral fats [41], therefore women present higher prevalences. The only non-significant interaction of sex and age concerned HDL cholesterol given its genetic component [42,43]. This particular dyslipidemia, highly prevalent in Mexico, affects nearly 50% of adults [44]. 

It is worth emphasizing two findings independent of the risk stemming from the visceral fats evaluated by WHtR. On one hand, women experienced a greater risk for total and high LDL-c after age 50 compared with men, a phenomenon that could be explained by the metabolic adaptations characteristic of menopause, the average age of which coincides with this decade [45]. On the other hand, men in their 40s experienced a greater risk for high triglycerides compared with women, possibly as a result of greater consumption of sugary drinks and alcohol [46,47]. 

In all interactions, we observed a decline in CVR indicators after age 70. This could be a result of loss of sensitivity of WHtR in the cutoff point (0.5) owing to the reduced height characteristic of aging [48,49,50]. A number of reports consolidate WHtR as a measure of adiposity and CVR factor in the older adult population; however, tipping points range from 0.5 to 0.6 [51,52,53], for which reason the cutoff point for WHtR in this study could be causing an underestimation of the real CVR for these age decades. In the case of HBP, risk was exponential in accordance with age. In the Mexican population, we observed that the prevalence of HBP increased with age and central adiposity [54], findings similar to those referred to by Macia et al. in Senegal [55]. 

Among the limitations of this study was the lack of agreement on the cutoff point used for measuring IR in the Mexican population. Existing studies have used cutoff points between 2.3 and 3.8 for adults from other countries [56,57,58,59]. For this study, we used a cutoff point of greater sensitivity (IR ≥ 2.6) in order to avoid failing to identify individuals with IR. Another limitation was the lack of agreement concerning the cutoff point for high WHtR in the Mexican population; for this reason, we performed a global analysis of ROC curves to ascertain which WHtR cutoff point demonstrated the greatest sensitivity and specificity for CVR indicators, finding a range of values between 0.52 and 0.60, being slightly higher in women. In comparing the results of the analysis utilizing the WHtR cutoff point of 0.5 and that derived from analyzing the ROC curves, we found similar results for CVR (data not shown). Finally, in order to provide information comparable with other studies, we utilized the cutoff point recommended internationally. 

The principal strengths of this analysis were its sample size and innovative role as the first-ever study to analyze the association between WHtR and CVR factors in the Mexican population. 

Beyond the traditional classifications based on BMI, the use of the WHtR can improve the sensitivity of efforts to identify subjects at risk. Evidence suggests that WHtR could be a preferable indicator for predicting short-term health risks compared with a combination of BMI and WC, given that some subjects with normal BMI and elevated WC are at risk for metabolic abnormalities [60]. It is worth noting that, in our analysis, more than half of adults classified as being of normal weight experienced high WHtR. 

## 5. Conclusions

High WHtR (≥0.5) was strongly associated with CVR factors in sampled adults. Age and sex formed differential patterns with respect to this risk, where women had higher CVR compared to men, regardless of their levels of visceral fat. This index can be used to predict adverse health outcomes for adults with greater sensitivity, and together with BMI, contributes to a more accurate diagnosis for cardiovascular disease in adults.

## Figures and Tables

**Figure 1 nutrients-11-01402-f001:**
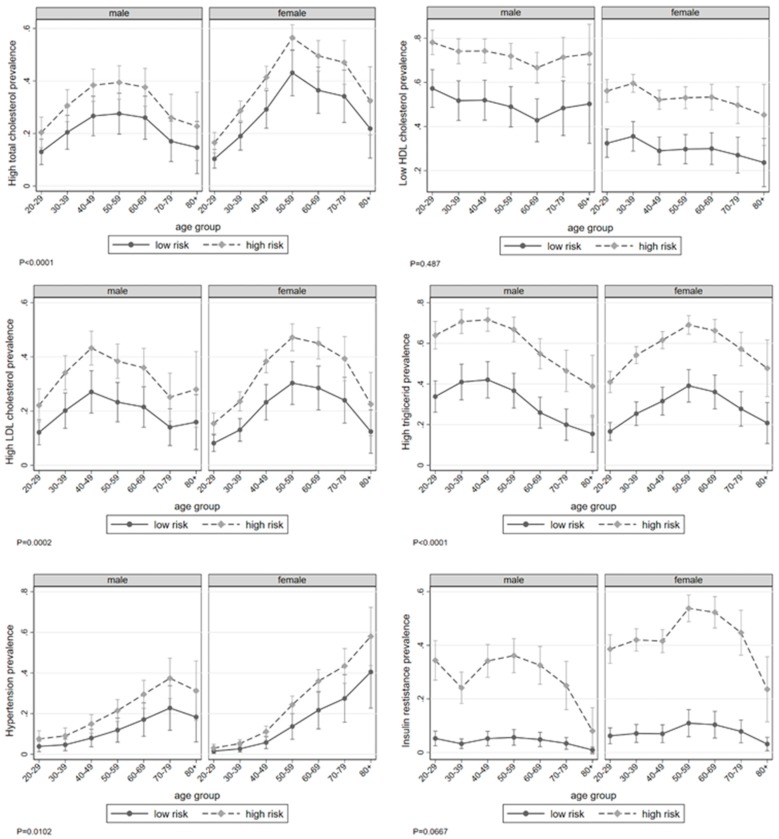
Age and sex interaction on prevalence rate for dyslipidemias, hypertension and insulin resistance by WHtR category for adults > 20 y, ENSANUT MC 2016, México. Low risk = WHtR < 0.5; High risk = WHtR ≥ 0.5.

**Table 1 nutrients-11-01402-t001:** Anthropometric and cardiovascular risk factors in adults over 20 years of age, by sex, ENSANUT MC 2016. Mexico, 2016.

Variables	Men	Women	Total
	Mean ± SD	Mean ± SD	Mean ± SD
	(*n* = 1225)	(*n* = 2325)	(*n* = 3550)
Age (years)	47.3 ± 16.6	45.4 ± 15.6 *	46.1 ± 16.0
Height (cm)	164.1 ± 7.7	152.1 ± 6.9 *	156.3 ± 9.2
Body weight (kg)	74.8 ± 15.9	67.5 ± 14.5	70.0 ± 15.4
BMI	27.5 ± 4.9	29.1 ± 5.5	28.5 ± 5.4
Waist circumference (cm)	96.2 ±13.2	95.0 ± 13.1*	95.4 ± 13.2
Risk at WHtR ¥ (%) 1	87.9	93.2	91.4
Lipidic profile			
Total Cholesterol (mg/dL)	184.4 ± 36.7	190.3 ± 40.7 *	188.2 ± 39.4
HDL Cholesterol (mg/dL)	36.2 ± 10.7	40.2 ± 10.3 *	38.9 ± 10.6
LDL Cholesterol (mg/dL)	110.2 ± 30.5	113.9 ± 32.2 *	112.7 ± 31.7
Trigrlicerid (mg/dL)	221.1 ± 167.6	192.0 ± 132.4 *	202.0 ± 146.2
Systolic pressure (mmHg)	125.0 ± 16.9	119.3 ± 20.1 *	121.0 ± 19.3
Diastolic pressure (mmHg)	73.7 ± 10.6	72.3 ± 10.6 *	72.8 ± 10.6
Glucose	104.8 ± 39.0	110.3 ± 48.5 *	108.0 ± 45.5
Insulin (uui/mL)	10.0 ± 11.5	12.0 ± 9.9 *	11.3 ± 10.5

¥ WHtR, waist-to-height ratio; * Different from men, t-Student *p* < 0.0001; 1 Prevalence of WHtR ≥0.5.

**Table 2 nutrients-11-01402-t002:** Lipid profile, blood pressure and insulin resistance in adults 20 years and older by category of ICE, ENSANUT MC 2016. Mexico, 2016.

Cardiovascular Risk Factors	WHtR¥ Category		
	No Risk	Risk	
	(*n* = 306)	(*n* = 3244)	*p*-Value *
	(< 0.5)	(≥ 0.5)	
Total Cholesterol (mg/dL)	172.7 ± 36.9	189.8 ± 39.4	<0.001
HDL Cholesterol (mg/dL)	43.3 ± 11.9	38.5 ± 10.4	<0.001
LDL Cholesterol 1 (mg/dL)	104.7 ± 29.6	113.5 ± 31.9	<0.001
Trigrlicerid 2 (mg/dL)	129.3 ± 74.3	208.7 ± 149.3	<0.001
Systolic pressure 3 (mmHg)	114.7 ± 17.2	121.9 ± 19.4	<0.001
Diastolic pressure 3 (mmHg)	68.7 ± 9.5	73.2 ± 10.7	<0.001
Glucose 4 (mg/dL)	93.9 ± 32.9	109.8 ± 46.3	<0.001
Insulin (uui/mL)	5.3 ± 3.8	11.9 ± 10.8	<0.001

¥WHtR, waist-to-height ratio; ***** Student’s t test by WHtR category; 1 No data 236; 2 No data 36; 3 No data 85; 4 No data 21.

**Table 3 nutrients-11-01402-t003:** Prevalence ratios by sex for dyslipidemia, hypertension and insulin resistance in adults over 20 years of age, by cardiovascular risk category assessed by ICE, ENSANUT MC 2016. Mexico, 2016.

	Men				Women			
	(*n* = 1225)				(*n* = 2325)			
Response Variable	WHtR ¥ Category				WHtR ¥ Category			
	No Risk	Risk	PR	95% CI	No Risk	Risk	PR	95% CI
	(< 0.5)	(≥ 0.5)			(< 0.5)	(≥ 0.5)		
	%	%			%	%		
High total Cholesterol 1 (mg/dL)	16.9	33.3	1.9	1.4–2.8	21.5	38.5	1.8	1.3–2.4
Low HDL Cholesterol 2 (mg/dL)	54.7	72.6	1.3	1.1–1.5	30.4	54.6	1.8	1.4–2.3
High LDL Cholesterol 3 (mg/dL)	14.9	35	2.3	1.6–3.5	17.1	33.5	1.9	1.4–2.8
High trigrlicerid 4 (mg/dL)	34.5	63.8	1.8	1.5–2.3	24	58.2	2.4	1.8–3.2
Hypertension 5	5.6	19.1	3.4	1.7–6.7	8.3	16.9	2.0	1.2–3.5
Insulin resistance 6 (uui/mL)	4.0	30.1	7.6	3.5–16.8	7.6	44.5	5.9	3.4–10.1

¥WHtR, waist-to-height ratio;1 Total Cholesterol ≥200 mg/dL; 2Low HDL Cholesterol <40 mg/dL; 3High LDL Cholesterol ≥130 mg/dL; 4High trigrlicerid ≥ 150 mg/dL; 5 Hypertension: systolic blood pressure ≥140 mmHg y/o diastolic blood pressure ≥90 mmHg; 6 Insulin resistance ≥2.6.

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
