# Peer review of "Association between High Waist-to-Height Ratio and Cardiovascular Risk among Adults Sampled by the 2016 Half-Way National Health and Nutrition Survey in Mexico (ENSANUT MC 2016)"

_nutrients, 2019, doi:10.3390/nu11061402_

Reviewer 1 Report

Research regarding this ethnicity group is much needed. I applaud you for bringing this information to the forefront. Just make sure to do a final spellcheck and grammar check just to make sure of appropriateness. However, I found that this information and the writing style was alright.

Author Response

Response 1: Thank you very much for your comments. We have made a review on spelling and grammar to the manuscript.

Reviewer 2 Report

 In the present study, the authors examined the association between a high WHtR and CVR indicators among Mexican adults canvassed by the 2016 Half-Way National Health and Nutrition Survey, and to examine the interaction of sex and age on this association.

Over 90% of participants had high WHtRs and were at greater risk for dyslipidemias, HBP and IR compared to those had low WHtRs. PR for men with high WHtRs were between 1.3 to 2.3 for dyslipidemias, 3.4 for high blood pressure (HBP) and 7.6 for Insulin Resistance (IR); among women were between 1.8 to 2.4 for dyslipidemias and HBP and 5.9 for IR. A high WHtR is associated with CVR factors in Mexican adults.

 The findings are important, however, I have several concerns.

 1. In general, obesity, IR, dyslipidemias and HBP are considered CVD risk factors. Therefore, the findings are not surprising even in the Mexican population. Authors should describe and discuss the originality and novelty of the present study in greater detail.

 2. Authors should describe the cutoff point for high WHtR in the Mexican men and women in the present study. What is the rationale of cutoff point between gender?

 3. WC measurements were taken on umbilical region? WC was measured with a flexible fiberglass tape with 1-mm precision. Authors should present the accuracy and reproducibility of the method.

 4. In the methods, blood specimens were drawn from the antecubital vein in vacuum tubes. Authors should add the timing of blood sampling and feeding condition.

 5. In Table 1, the authors should add the data of body weight and BMI.

 6. In Figure 1, age and sex interaction on prevalence rate for dyslipidemias, hypertension and insulin resistance by WHtR category. The distribution of the prevalence rate is different between several markers. Authors should discuss the point including gender differences.

Author Response

 1. In general, obesity, IR, dyslipidemias and HBP are considered CVD risk factors. Therefore, the findings are not surprising even in the Mexican population. Authors should describe and discuss the originality and novelty of the present study in greater detail.

 Response 1: We modified the wording in the discussion section. Lines 176-178.

 2. Authors should describe the cutoff point for high WHtR in the Mexican men and women in the present study. What is the rationale of cutoff point between gender?

 Response 2: We added information that highlights the importance of the findings in women. Lineas193-194, 218.

3. WC measurements were taken on umbilical region? WC was measured with a flexible fiberglass tape with 1-mm precision. Authors should present the accuracy and reproducibility of the method.

Response 3: Waist circumference was measured twice at the level of the navel, according to the standardized technique of Lohman et al. The difference between both measures should not exceed 5mm. This description has been added to the methods section. Lines 80-82.

 4. In the methods, blood specimens were drawn from the antecubital vein in vacuum tubes. Authors should add the timing of blood sampling and feeding condition.

 Response 4: Samples were obtained considering at least 8 hours of fasting, during the morning of the interview. This information was included in the manuscript. Lines 86, 91-92.

5. In Table 1, the authors should add the data of body weight and BMI.

Response 5: The information has been added in table 1

 6. In Figure 1, age and sex interaction on prevalence rate for dyslipidemias, hypertension and insulin resistance by WHtR category. The distribution of the prevalence rate is different between several markers. Authors should discuss the point including gender differences.

Response 5: Information that highlights that women have a higher prevalence than men was included. Lines 193-194.

Reviewer 3 Report

Waist-to-height-ratio has been shown as a useful tool for assessing obesity and other metabolic syndromes. In this manuscript, the authors analyzed more than 3500 cases in Mexico and got the conclusion that high WHtR was strongly associated with CVR factors. There are a few concerns that need to be addressed. The comments on this review are as followed:

1. The authors analyzed data from 3550 adults who are older than 20 years old. As it was reported the association of WHtR and CVD risk differed among age groups due to different whole-body metabolism, they could split the data to different year cohort to avoid misleading results caused by age difference.

2. In the Materials and Methods, the authors didn’t show the information about if participants have any history of heart disease, stroke or cancer.

3. In the discussion, it was claimed that WHtR is a preferable indicator compared with a combination of BMI and WC, the authors could compare the association of WHtR with CVD and the association of BMI and WC with CVD to further confirm their conclusion.

4. In line 180, please check if there’s a mistake.

Author Response

1. The authors analyzed data from 3550 adults who are older than 20 years old. As it was reported the association of WHtR and CVD risk differed among age groups due to different whole-body metabolism, they could split the data to different year cohort to avoid misleading results caused by age difference.

Response 1: Thanks for the comment, however, Figure 1 shows the sex and age group interaction, where risk factors have different patterns by decades of age.

2. In the Materials and Methods, the authors didn’t show the information about if participants have any history of heart disease, stroke or cancer.

Response 2: Thanks for the comment. Regarding the information on health of participants, less than 2% of the sample suffered a heart attack, which do not affect the estimates in our work, we highlight the relationship of the waist-to-height-ratio as a proxy to the CVD Risk. There is no information on the history of cancer for the study participants.

3. In the discussion, it was claimed that WHtR is a preferable indicator compared with a combination of BMI and WC, the authors could compare the association of WHtR with CVD and the association of BMI and WC with CVD to further confirm their conclusion.

Response 3: It is a very interesting comment, however, our work does not aim to confirm the superiority of the waist-to-height-ratio on the BMI and the waist circumference on its association with cardiovascular risk, the meaning is to show the association of the waist-to-height-ratio with factors of cardiovascular risk in Mexican adults and describe the interaction of sex and age on these relationships.  We will make a change in the writing of the manuscript which mentions this assertion.

4. In line 180, please check if there’s a mistake.

Response 4: It is a mistake due to a crossed reference. It was corrected in the document. Lines 83 and 182

Round  2

Reviewer 2 Report

The addition and correction have improved the manuscript.

I have no further concern. 

Reviewer 3 Report

The authors have adequately addressed my concerns in the revised manuscript.